# Subthalamic beta-targeted neurofeedback speeds up movement initiation but increases tremor in Parkinsonian patients

Shenghong He[1,2], Abteen Mostofi[3], Emilie Syed[1], Flavie Torrecillos[1,2], Gerd Tinkhauser[1,4], Petra Fischer[1,2], Alek Pogosyan[1,2], Harutomo Hasegawa[5], Yuanqing Li[6], Keyoumars Ashkan[5], Erlick Pereira[3], Peter Brown[1,2]\*, Huiling Tan[1,2]\*

[1]MRC Brain Network Dynamics Unit at the University of Oxford, Oxford, United Kingdom; [2]Nuffield Department of Clinical Neurosciences, University of Oxford, Oxford, United Kingdom; [3]Neurosciences Research Centre, Molecular and Clinical Sciences Research Institute, St George's University of London, London, United Kingdom; [4]Department of Neurology, Bern University Hospital and University of Bern, Bern, Switzerland; [5]Department of Neurosurgery, King's College Hospital NHS Foundation Trust, King's Health Partners, London, United Kingdom; [6]School of Automation Science and Engineering, South China University of Technology, Guangzhou, China

\*For correspondence:
peter.brown@ndcn.ox.ac.uk (PB);
huiling.tan@ndcn.ox.ac.uk (HT)

**Abstract** Previous studies have explored neurofeedback training for Parkinsonian patients to suppress beta oscillations in the subthalamic nucleus (STN). However, its impacts on movements and Parkinsonian tremor are unclear. We developed a neurofeedback paradigm targeting STN beta bursts and investigated whether neurofeedback training could improve motor initiation in Parkinson's disease compared to passive observation. Our task additionally allowed us to test which endogenous changes in oscillatory STN activities are associated with trial-to-trial motor performance. Neurofeedback training reduced beta synchrony and increased gamma activity within the STN, and reduced beta band coupling between the STN and motor cortex. These changes were accompanied by reduced reaction times in subsequently cued movements. However, in Parkinsonian patients with pre-existing symptoms of tremor, successful volitional beta suppression was associated with an amplification of tremor which correlated with theta band activity in STN local field potentials, suggesting an additional cross-frequency interaction between STN beta and theta activities.

## Introduction

Enhanced synchronization of neural activity in the beta band (13–30 Hz) has been consistently observed in the subthalamic nucleus (STN) in patients with Parkinson's disease (**Kühn et al., 2009**; **Neumann et al., 2016**). Synchrony in this frequency band takes the form of short-lived bursts of different durations and amplitudes (**Tinkhauser et al., 2017a**; **Tinkhauser et al., 2017b**). The occurrence rate of longer beta bursts with large amplitude positively correlates with motor impairment (**Tinkhauser et al., 2017a**; **Tinkhauser et al., 2020**; **Torrecillos et al., 2018**). Closed-loop deep brain stimulation (DBS), which selectively truncates long duration beta bursts, can achieve clinical improvement that is at least as good as that with conventional continuous DBS in acute trials (**Little et al., 2013**; **Little et al., 2016**). These studies highlight the importance of modulating the temporal dynamics of beta activity in the STN for the treatment of Parkinson's disease.

A better understanding of the electrophysiological biomarkers underlying symptoms of bradykinesia and rigidity in Parkinson's disease has motivated the use of neurofeedback as a therapeutic

technique for the disease (*Esmail and Linden, 2014*; *Fukuma et al., 2018*; *Carney, 2019*). In neurofeedback training, neural activities were recorded and quantified in real-time and provided to the participant for the purpose of self-regulation (*Sitaram et al., 2017*). Parkinsonian patients have been shown to be capable of voluntarily regulating STN beta-band power measured from electrodes implanted for DBS (*Carney, 2019*; *He et al., 2019*). However, it is still not clear whether modulating beta oscillations in STN through neurofeedback training can lead to changes in motor performance in patients with Parkinson's disease (*Subramanian et al., 2011*; *Erickson-Davis et al., 2012*). Additionally, previous studies have not specifically targeted bursts of prolonged beta activity, nor considered any additional effects of beta-targeted neurofeedback training on tremor.

Tremor is another cardinal symptom of Parkinson's disease. Its pathophysiology remains poorly understood, but some recent studies indicate that the pattern of neural activities related to Parkinsonian tremor can be very different from those related to bradykinesia and rigidity. For example, reduced activities in the beta band and increases in power in the tremor frequency band, corresponding to the theta band (3–7 Hz), in the STN, as well as reduced basal ganglia-cortical coherence in the beta frequency band have been observed during the presence of resting tremor in Parkinson's disease (*Hirschmann et al., 2013*; *Qasim et al., 2016*; *Asch et al., 2020*). Moreover, one in five patients shows resurgence of tremor if DBS is only switched on when STN beta activity is high (*Little and Brown, 2020*). These observations raise the possibility that neurofeedback training that suppresses beta oscillations in the STN may not improve, even worsen, resting tremor in Parkinsonian patients.

In this study, we adopted a sequential neurofeedback-behavior task to test whether modulating beta oscillations in the STN through neurofeedback training can lead to changes in motor initiation and whether the endogenous suppression of STN beta band activities increases resting tremor in Parkinson's disease. Similar experimental designs have helped to shed light on the relationship between neural activity and behavior (*McFarland et al., 2015*; *Khanna and Carmena, 2017*). In a recent study, we showed that healthy young participants can indeed suppress cortical beta measured using electroencephalogram (EEG) with veritable neurofeedback better than sham feedback (*He et al., 2020*). In the paradigm of the current study, a cued finger pinch movement followed a neurofeedback phase during which the position of a visual cue was controlled by suppressing high amplitude beta bursts in activities measured by DBS electrodes implanted in the STN. The endogenous changes in subthalamic activities induced by neurofeedback training also allow us to investigate the relationship between subthalamic activities and motor performance, as well as the severity of tremor on a trial-to-trial basis in patients with Parkinson's disease.

## Results

### Neurofeedback control was achieved within 1 day of training

Twelve Parkinsonian patients, who underwent bilateral implantation of DBS electrodes targeting the motor area of the STN, participated in this study during the time when the DBS leads were temporarily externalized. The position of a basketball displayed on a monitor was used as the visual feedback about the incidence of beta bursts detected in STN local field potentials (LFPs) (*Figure 1A*). The bipolar LFP channel and the peak frequency bands (5 Hz width) with the largest movement-related changes between 13 and 30 Hz were selected to drive the visual feedback for each hemisphere (*Figure 2*). Specifically, the average power in the selected beta frequency band over each 500 ms time window was used as a neurofeedback signal to control the vertical position of the basketball. In real-time, we assumed that a beta burst was detected when the average beta power within the past 500 ms time window exceeded a predefined threshold, which would result in a drop of the basketball. The patient details and patient-specific beta frequency bands are presented in *Table 1*. Each patient completed at least four sessions of the task with 10 trials in the 'Training' condition and 10 trials in the 'No Training' condition in each session with two hands separately (*Figure 1B*). The participants were asked to keep the position of the basketball high (corresponding to reduced beta bursts) during the neurofeedback phase in the 'Training' condition. In the 'No Training' control condition, they were asked to pay attention to the position of the basketball without trying to control it, though the ball was also moving toward the right as in the 'Training' condition, and the vertical position was controlled by the natural ongoing variations in beta activity. The average

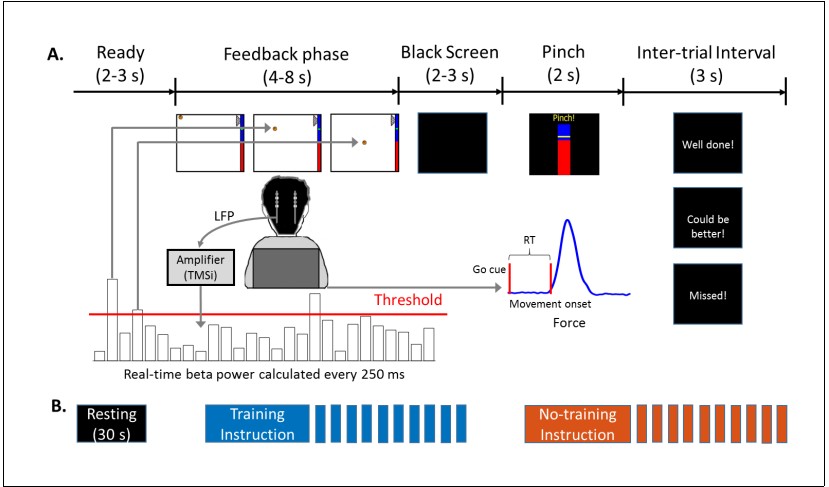

**Figure 1.** Experimental protocol. (**A**) Timeline of one individual trial. Each trial consisted of a neurofeedback phase followed by a cued pinch movement. After the finger pinch motor task, a message was displayed ('Well done!' or 'Could be better!') depending on whether the reaction time of the previous movement was shorter or longer than 800 ms. If movement onset was not detected within 2 s after the Go cue, the message 'Missed!' was displayed. (**B**) Timeline of one experimental session which consisted of 30 s of resting, and one block of 10 trials in the 'Training' condition (when participants were instructed to keep the basketball floating) and one block of 10 trials in the 'No Training' condition (when the participants were instructed to just pay attention to the movement of the basketball). The order of the 'Training' and 'No raining' blocks was randomized across sessions. At the beginning of each session the threshold was recalculated based on recordings made at rest.

final basketball position in the vertical axis, which reflected the performance of neurofeedback control, was calculated for each tested hemisphere in each experimental condition. Paired *t*-test showed that the final basketball position was higher in the 'Training' condition compared to the 'No Training' condition ($t_{20}$ = 4.6054, p = 0.0002, *Figure 3A*), and this was not consequent on physical movement that was monitored by EMGs attached to both forearms of the participants (*Figure 3B*).

## Neurofeedback training reduced beta oscillations in STN LFPs and reduced beta band synchrony between the conditioned STN and ipsilateral motor cortex compared to a passive observation task

Compared to the 'ready' period, activity in STN was reduced over a broad frequency band (7–30 Hz) during the neurofeedback phase in the 'Training' condition (shown in *Figure 3C*), similar to the

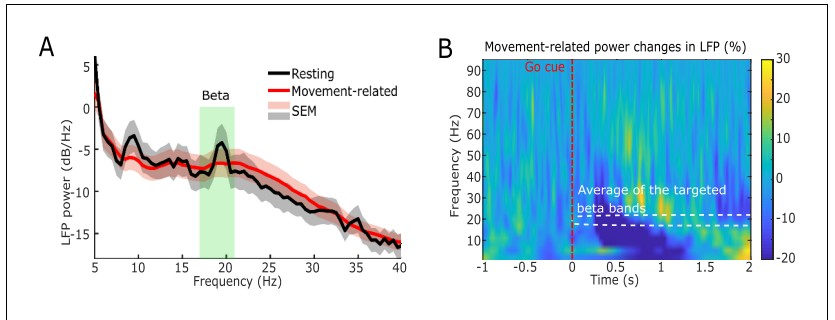

**Figure 2.** Power spectra of the neurofeedback-targeted subthalamic nucleus (STN) local field potential (LFP) signals averaged across 21 hemispheres. (**A**) Resting (black) and movement-related (red) power spectral density in STN LFP recorded during the calibration procedure. The green shaded area indicates the average of the targeted beta frequency bands. (**B**) Group average time-frequency power spectra locked to the Go cue (red dashed line) which prompted a finger pinch movement. The white dashed rectangle indicates the average targeted beta band. The blue color displays a decrease in power relative to the pre-cue baseline (expressed as percentage change).

**Table 1.** Patients' details.

| Patient | G | Age (yr) | DD (yr) | U off | U on | DBS lead | Selected contact (L/R) | Beta peak (L/R Hz) | Predominant symptom(s) before surgery |
|---|---|---|---|---|---|---|---|---|---|
| 1 | M | 48 | 17 | 71 | 37 | Bost | L03/R03 | 15/15 | Tremor |
| 2[a] | M | 66 | 15 | 57 | 34 | Medt | L23/R01 | 20/20 | Mixed |
| 3[a] | F | 70 | 20 | 54 | 19 | Medt | L01/R23 | 20/20 | Akinetic-rigid, tremor |
| 4 | M | 69 | 17 | 37 | 18.5 | Medt | L23/R23 | 21/20 | Akinetic-rigid, tremor |
| 5 | F | 66 | 10 | 53 | 30 | Bost | L01/R01 | 15/15 | Akinetic-rigid |
| 6[b] | M | 65 | 5 | 34 | 16 | Medt | L01/R23 | 15/25 | Akinetic-rigid |
| 7[a,b] | M | 61 | 9 | 33 | 12 | Bost | L01/R23 | 20/22 | Tremor |
| 8[c] | M | 49 | 8 | 45 | 34 | Bost | L01 | 15 | Tremor |
| 9[c] | F | 57 | 6 | 48 | 19 | Bost | L23 | 19 | Mixed |
| 10[b] | M | 51 | 12 | 27 | 13 | Bost | L23/R23 | 22/21 | Akinetic-rigid |
| 11[a,b] | M | 67 | 6 | N/A | N/A | Bost | L23/R23 | 19/19 | Tremor |
| 12[a,c] | F | 75 | 7 | 36 | 19 | Medt | R12 | 18 | Tremor, bradykinesia, freezing |
| Mean | - | 62 | 11 | 45 | 22.9 | - | - | 18.9 | - |
| SEM | - | 8.8 | 5.1 | 13.1 | 9.1 | - | - | 0.6 | - |

Patients 2, 3, 7, and 11 ([a]) had tremor during the experiment. Patients 6, 7, 10, and 11 ([b]) performed the test on two consecutive days. Patient 8, 9, and 12 ([c]) were only recorded on one side. G = gender; yr = year; U Off/On = UPDRS Off/On; DBS = deep brain stimulation; L/R = left/right; SEM = standard error of the mean; N/A = unknown; Bost = Vercise Cartesia Directional Lead, Boston Scientific; Medt = Quadripolar Macroelectrode, Model 3389, Medtronic.

actual movement related modulation shown in *Figure 2B*. A paired *t*-test confirmed a significant effect of neurofeedback in facilitating beta suppression in terms of the average normalized power in the selected beta bands ($t_{20} = -3.6975$, $p = 0.0014$, *Figure 4A*). The difference in the normalized beta power between the 'Training' and 'No Training' conditions correlated positively with the percentage change in the beta power during real movement ($r = 0.5896$, $p = 0.0057$, Pearson's correlation, *Figure 3D*). The neurofeedback training also led to reduced accumulated beta burst duration in the STN LFPs determined as percentage of time with beta amplitude being over the predefined threshold ($t_{20} = -4.7415$, $p = 0.0001$, $17.40 \pm 1.44\%$ compared to $22.43 \pm 1.85\%$, mean ± SEM, *Figure 4B*), a reduced average burst duration ($t_{20} = -3.9428$, $p = 0.0008$, $319.6 \pm 19.3$ ms compared to $377.2 \pm 21.5$ ms, *Figure 4C*), and a reduced number of bursts per second ($t_{20} = -4.8536$, $p = 0.0001$, $0.446 \pm 0.030$ compared to $0.531 \pm 0.033$, *Figure 4D*). The bursts with durations longer than 400 ms were reduced more consistently compared with the shorter bursts (*Figure 4—figure supplement 1*). In addition, we observed an increase in the broad gamma frequency band (55–95 Hz) in the STN LFPs ($t_{20} = 3.4899$, $p = 0.0023$, *Figure 5A*).

There was no significant change in the 'Beta−8Hz' (centered between 9.4 and 13.4 Hz, *Figure 5B*) or higher frequency band ('Beta+8' [centered between 25.4 and 29.4 Hz], *Figure 5C*).

Although there was a trend of reduction in the average normalized beta power and beta burst characteristics in the EEG recorded over the ipsilateral motor cortex, the changes were not significant or did not survive multiple comparison correction (*Figure 4D–H*). There was no significant change in the gamma activities in the EEG measured over the motor cortex ($z = 0.7821$, $p = 0.4342$).

The phase synchrony index ($t_{20} = -2.5462$, $p = 0.0192$, *Figure 4I*) and spectral coherence ($z = -3.1803$, $p = 0.0015$, *Figure 4J*) between the conditioned STN and ipsilateral motor cortex were significantly reduced in the beta band in the 'Training' condition compared with the 'No Training' condition, and this change did not happen in other frequency bands ('Beta−8' or 'Beta+8').

## Carry-over effect of neurofeedback training

There was a sustained carry-over effect of neurofeedback training over the short time window (~2 s) after the neurofeedback phase when a black screen was presented before the Go cue. The average normalized beta power ($k = 0.6050 \pm 0.0241$, $p < 0.0001$), accumulated beta burst duration ($k = 0.0892 \pm 0.0144$, $p < 0.0001$), and normalized gamma power ($k = 0.9617 \pm 0.0073$, $p < 0.0001$)

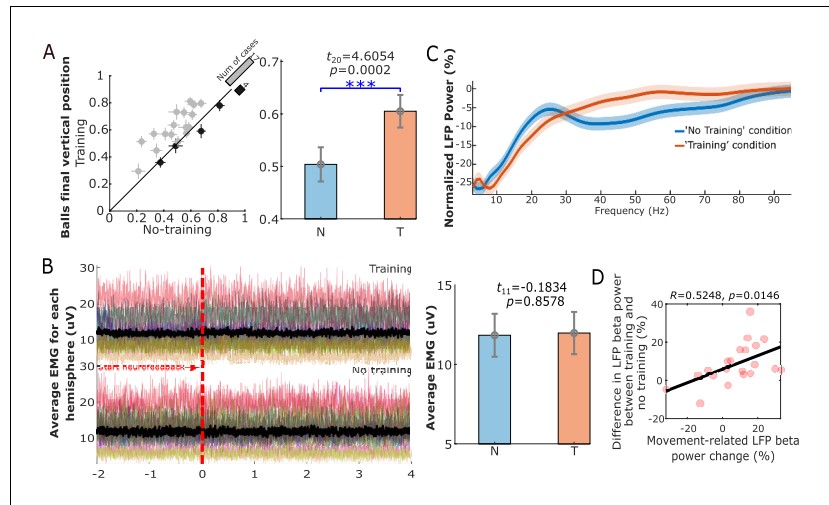

**Figure 3.** Neurofeedback training performance. (**A**) The final vertical position of the basketball for each individual hemisphere (left) and group-averaged balls' final vertical positions (mean ± SEM) in the 'Training' (T) and 'No Training' (N) conditions (right). The dots with crosses indicate the means and cross-trial SEMs for each tested hemisphere. The gray and dark-shaded dots indicate higher measurement in the 'Training' and 'No Training' conditions, respectively. The bar on the diagonal refers to the number of cases with higher measurement in each condition. The error bar plots on the right show the mean and SEM across all tested hemispheres in different conditions. (**B**) There was no significant difference between the rectified EMG amplitude during the neurofeedback phase in the 'Training' and 'No Training' conditions. Different colors on the left indicate the average EMGs for different hands contralateral to the tested hemispheres. The black line indicates the averaged EMG traces across hands in different conditions. The error bar plots on the right show the mean and SEM during the neurofeedback phase across hands in different conditions. (**C**) Group-averaged power spectra of the targeted STN LFP signals (normalized against the pre-cue resting period) in the 'Training' (orange) and 'No Training' (blue) conditions for different frequencies. Solid lines and the shaded areas show the average and SEM across all tested hemispheres. (**D**) The reduced beta power by neurofeedback training positively correlated with the movement-related power changes. Each pink dot indicates a hemisphere. ***p < 0.001.

during the 2 s pre-Go cue were positively correlated with the average normalized beta power, beta burst duration, and normalized gamma power during the 4 s feedback phase, respectively, as identified by the generalized linear mixed effects (GLME) modeling using the measurements during the 2 s pre-Cue and 4 s feedback phase as the dependent variables and predictors, respectively. If we replaced the predictor by the experimental condition ('Training' or 'No Training') in the models, the results revealed that the average beta power ($k = -0.2523 \pm 0.0769$, p = 0.0011) and accumulated beta burst duration ($k = -0.0601 \pm 0.0172$, p = 0.0005) during the 2 s pre-Go cue were significantly reduced in the 'Training' condition compared to the 'No Training' condition. In contrast, the average gamma power during the 2 s pre-Go cue were significantly increased ($k = 0.0781 \pm 0.0296$, p = 0.0083) in the 'Training' condition compared to the 'No Training' condition.

## Neurofeedback training improved reaction time in subsequently cued movements

The reaction time in response to the Go cue was significantly reduced in the 'Training' condition compared with the 'No Training' condition (487.4 ± 29.7 ms compared to 510.9 ± 32.3 ms, $t_{20}$ = −2.7518, p = 0.0123, paired t-test, *Figure 6A*). *Figure 6B* shows an example of the recorded left-hand pinch force in the 'Training' and 'No Training' conditions from Patient 12.

GLME modeling was used to investigate the relationship between the reaction time and the STN LFP activities in the beta ($\beta$) and gamma ($\gamma$) frequency bands considering all valid trials for both the 'Training' and 'No Training' conditions across all tested hemispheres. We focused on the neural activities during the 2 s window before the Go-cue when the visual neurofeedback was no longer presented. When STN average beta power or beta burst characteristics (average burst duration, accumulated burst duration) during the 2s before the Go-cue were used as the only predictor in

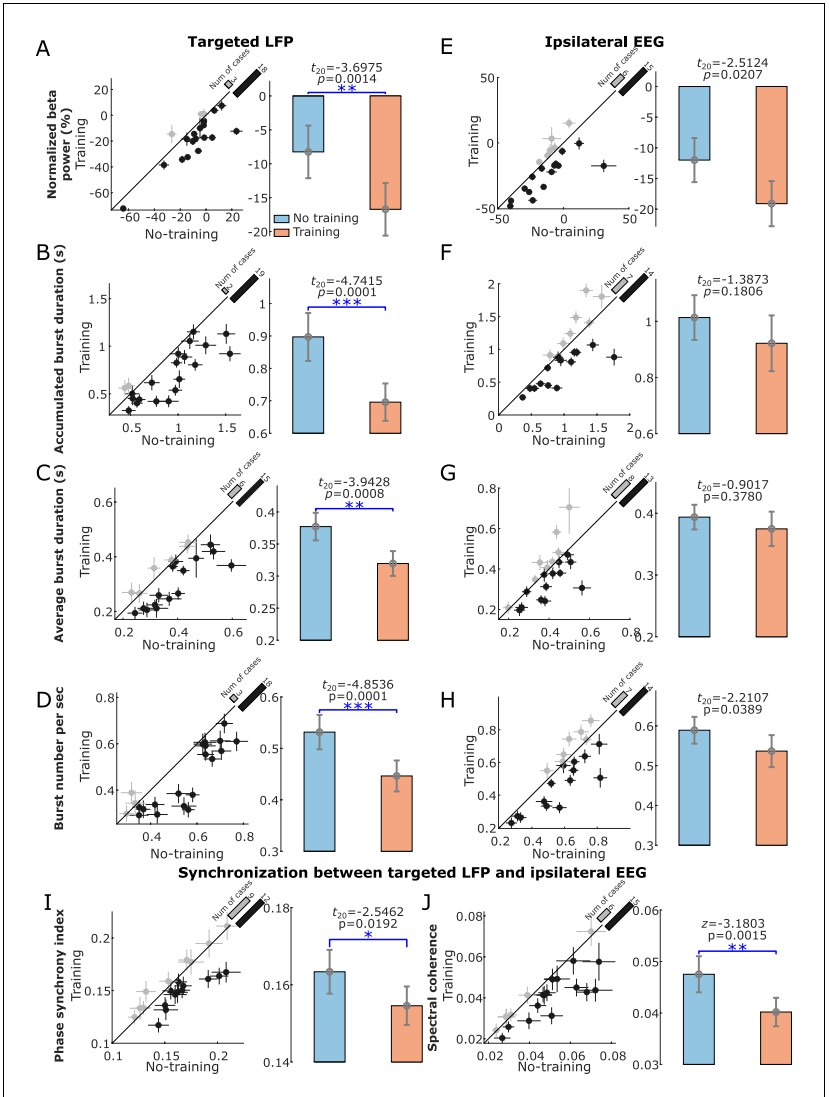

**Figure 4.** Normalized beta power and burst characteristics in the targeted subthalamic nucleus (STN) local field potential (LFP) and electroencephalogram (EEG) from ipsilateral motor cortex. (**A–D**) Normalized beta power (**A**), total burst duration (**B**), average burst duration (**C**), and number of beta bursts per second (**D**) in the STN LFP were all significantly reduced in the 'Training' condition compared to the 'No Training' condition. (**E–H**) The same for EEG from ipsilateral motor cortex. (**I and J**) The phase synchrony index (**I**) and spectral coherence (**J**) between STN and ipsilateral motor cortex were significantly reduced in 'Training' condition compared with 'No Training' condition. The dots with crosses indicate the means and cross-trial SEMs for each tested hemisphere. The gray and dark-shaded dots indicate higher measurement in the 'Training' and 'No Training' conditions, respectively. The bar on the diagonal refers to the number of cases with higher measurement in each condition. The error bar plots on the right show the mean and SEM across all tested hemispheres in different conditions; *p < 0.05, **p < 0.01/4 in (**A**) and (**C**), **p < 0.01 in (**J**), ***p < 0.001/4; Beta indicates hemisphere-specific beta band. The online version of this article includes the following figure supplement(s) for figure 4:

**Figure supplement 1.** Distribution profiles of the beta bursts of different durations during the 4 s feedback phase in the 'Training' (orange) and 'No Training' (blue) conditions.

separate models, all of them significantly contributed to the prediction of reaction time (Models 1–5, *Table 2*). We then used the model of $RT \sim k_1 \times T\,or\,N + k_2 \times \beta + k_3 \times \gamma + k_4 \times \alpha + 1|SubID$ (Model 6) to evaluate if activities in broad band gamma ($\gamma$) and alpha ($\alpha$) frequency bands also contributed to the prediction of reaction time. In the latter model, only average beta power ($\beta$) was used so as to keep the unit of beta similar to that of the other frequency bands used. This model confirmed the

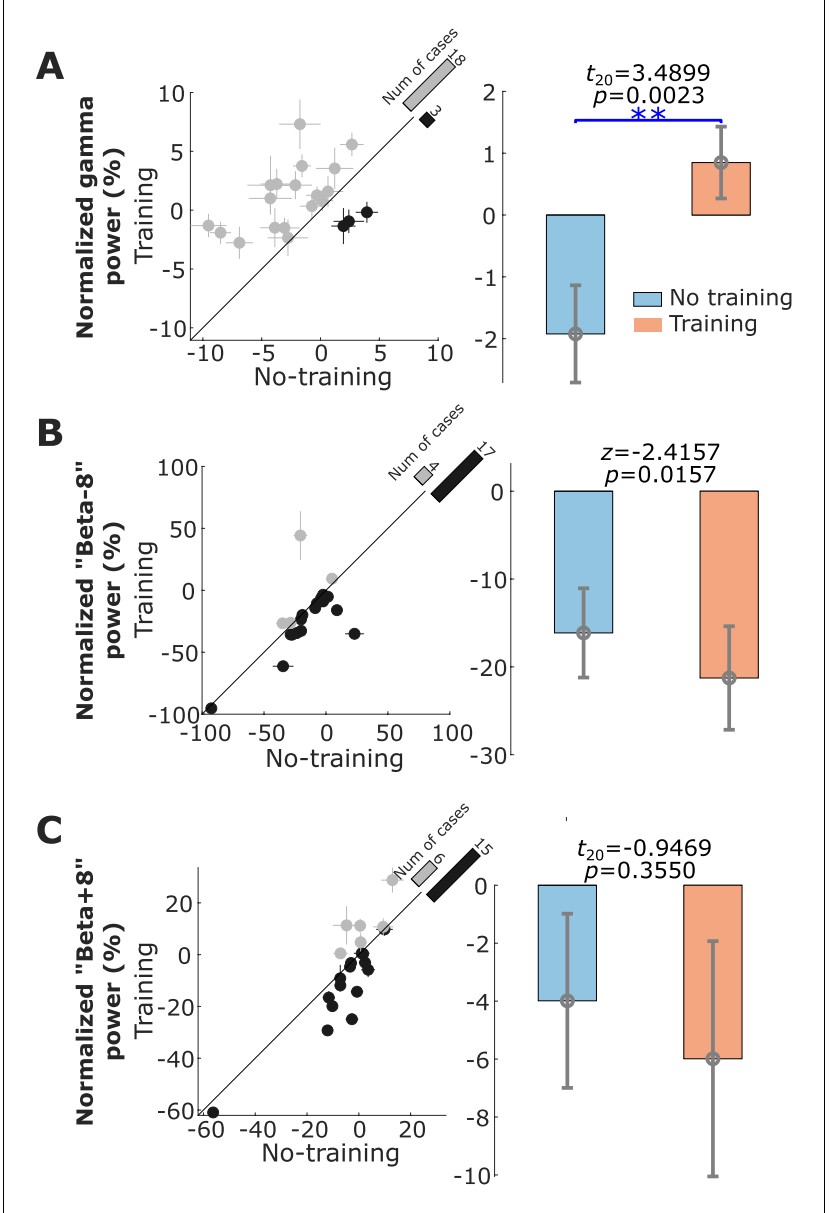

**Figure 5.** Normalized power in the gamma, 'Beta−8', and 'Beta+8' frequency bands associated with neurofeedback training in the targeted subthalamic nucleus (STN) local field potential (LFP). (**A**) The average normalized gamma (55–95 Hz) power in the STN LFP was significantly increased in the 'Training' condition compared with the 'No Training' condition. (**B** and **C**) There was no significant change in the power percentage change in the 'Beta−8' frequency band and the 'Beta+8' frequency band between the 'Training' and 'No Training' conditions. The dots with crosses indicate the means and cross-trial SEMs for each tested hemisphere. The gray and dark-shaded dots indicate higher measurement in the 'Training' and 'No Training' conditions, respectively. The bar on the diagonal refers to the number of cases with higher measurement in each condition. The error bar plots on the right show the mean and SEM across all tested hemispheres in different conditions; **p < 0.01; Beta indicates hemisphere-specific beta band.

significant effect of beta-targeted neurofeedback training (i.e., whether patients were in the 'Training' or 'No Training' condition) in reducing reaction time ($T\,or\,N$: $k_1$ = −0.0154 ± 0.0071, p = 0.0297), and of a significant positive effect of the beta band power ($\beta$: $k_2$ = 0.0061 ± 0.0020, p = 0.0017) and negative effect of gamma band power ($\gamma$:$k_3$ = −0.0085 ± 0.0026, p = 0.0014) in the STN LFPs over the 2 s before the Go-cue. There was no significant effect of alpha band activity on

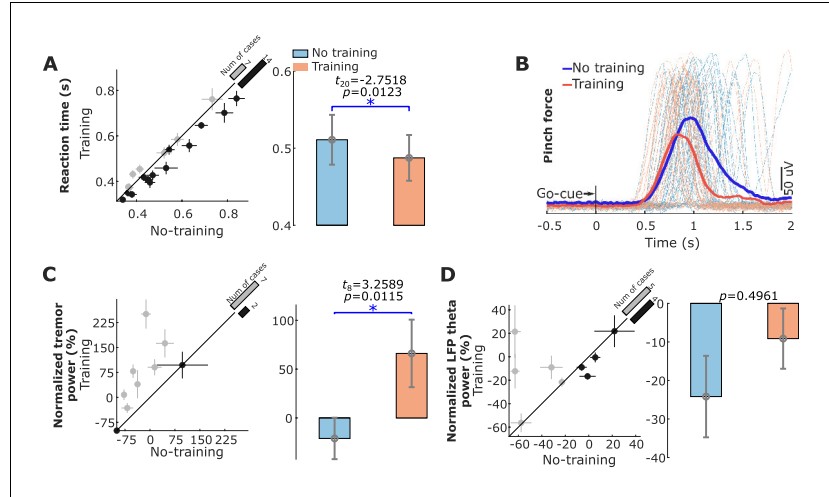

**Figure 6.** Behavioral changes (reaction time and tremor) associated with neurofeedback training. (**A**) The reaction time for each individual hemisphere (left) and group-averaged reaction time in the 'Training' and 'No Training' conditions (right). (**B**) Recorded left-hand pinch force in the 'Training' (red) and 'No Training' (blue) conditions for each individual trial (dashed line) and the trial-averaged curves (solid lines) from Patient 12. (**C**) Normalized tremor power quantified based on measurements from the accelerometer in the 'Training' and 'No Training' conditions for the nine hemispheres that displayed contralateral tremor during the experiment. (**D**) Normalized power in the tremor frequency band in the subthalamic nucleus local field potential for the nine hemispheres that displayed contralateral tremor during the experiment. * indicates significance after correction for multiple comparison p < 0.0167.

The online version of this article includes the following figure supplement(s) for figure 6:

**Figure supplement 1.** No significant difference in the reaction time, normalized gamma power, and normalized tremor power between trails from 'Training' and 'No Training' conditions with similar normalized beta power.

**Figure supplement 2.** Subthalamic nucleus (STN) local field potential (LFP) theta power positively correlated with tremor power.

reaction time ($\alpha : k_4$ = 0.0029 ± 0.0022, p = 0.1948). Overall, around 20% of the variance in the reaction time was being explained by the model (Model 6, $R^2$ = 0.2072, *Table 2*). The significant negative $k_1$ showed that there was an effect of 'Training' in reducing the reaction time, which cannot be explained by changes in the beta or gamma band power. The positive sign of $k_2$ and negative sign of $k_3$ indicate that reduced STN beta band power and increased gamma band power over the 2 s before the Go-cue predicted faster reaction time. In addition, we selected a subgroup (75%) of trials from the 'Training' and 'No Training' conditions that have similar normalized beta power (*Figure 6—figure supplement 1A*), and tested the differences in reaction time and normalized gamma power. The results showed no significant difference in the RT ($t_{20}$ = −0.4374, p = 0.6665, *Figure 6—figure supplement 1B*) nor in the normalized gamma power (z = −0.8168, p = 0.4140, *Figure 6—figure supplement 1C*) between the selected trials from the 'Training' and 'No Training' conditions but with matched normalized beta power. Overall these analyses suggest that beta modulation during neurofeedback training does contribute to the changes in RT, even though other condition factors (e.g., cognitive requirement) may also contribute to the observed difference in the RT between the 'Training' and 'No Training' conditions.

When the EEG beta band and alpha band activities, and the experimental condition were considered as the only predictors in the model, the EEG beta band activity also contributed to the prediction of reaction time (k = 0.0067 ± 0.0024, p = 0.0058, Model 8, *Table 2*), consistent with previous findings in young healthy participants (*He et al., 2020*). However, when EEG beta, STN beta, and STN gamma were considered together in one model, only STN beta and STN gamma significantly contributed to the prediction of reaction time (Model 9, *Table 2*).

**Table 2.** Generalized linear mixed effects modeling details.

| ID | Model | Akaike's information criterion (AIC) | k-Value | p-Value | $R^2$ |
|----|-------|-------------------------------------|---------|---------|-------|
| 1 | $RT \sim 1 + k \times T\,or\,N + 1|SubID$ | −1201.4 | $k = -0.0158 \pm 0.0072$ | $p = 0.0278$ | 0.1893 |
| 2 | $RT \sim 1 + k \times \beta LFP + 1|SubID$ | −1194.6 | $k = 0.0061 \pm 0.0019$ | $p = 0.0011$ | 0.1912 |
| 3 | $RT \sim 1 + k \times Dur1LFP + 1|SubID$ | −1189.5 | $k = 0.0284 \pm 0.0092$ | $p = 0.0021$ | 0.1897 |
| 4 | $RT \sim 1 + k \times Dur2LFP + 1|SubID$ | −1182.4 | $k = 0.0274 \pm 0.0136$ | $p = 0.0436$ | 0.1869 |
| 5 | $RT \sim 1 + k \times NumLFP + 1|SubID$ | −1190 | $k = 0.0231 \pm 0.0086$ | $p = 0.0074$ | 0.1888 |
| 6 | $RT \sim 1 + k_1 \times T\,or\,N + k_2 \times \beta LFP + k_3 \times \gamma LFP + k_4 \times \alpha LFP + 1|SubID$ | −1236.5 | $k_1 = -0.0152 \pm 0.0071$ <br> $k_2 = 0.0069 \pm 0.0020$ <br> $k_3 = -0.0010 \pm 0.0024$ <br> $k_4 = 0.0003 \pm 0.0013$ | $p_1 = 0.0316$ <br> $p_2 = 0.0008$ <br> $p_3 = 0.00003$ <br> $p_4 = 0.8365$ | 0.2072 |
| 7 | $RT \sim 1 + k \times \beta EEG + 1|SubID$ | −1195.7 | $k = 0.0074 \pm 0.0019$ | $p = 0.0001$ | 0.1924 |
| 8 | $RT \sim 1 + k_1 \times T\,or\,N + k_2 \times \beta EEG + k_3 \times \alpha EEG + 1|SubID$ | −1218.1 | $k_1 = -0.0158 \pm 0.0071$ <br> $k_2 = 0.0067 \pm 0.0024$ <br> $k_3 = 0.0007 \pm 0.0016$ | $p_1 = 0.0276$ <br> $p_2 = 0.0058$ <br> $p_3 = 0.6469$ | 0.1965 |
| 9 | $RT \sim 1 + k_1 \times T\,or\,N + k_2 \times \beta LFP + k_3 \times \gamma LFP + k_4 \times \beta EEG + 1|SubID$ | −1236.6 | $k_1 = -0.0154 \pm 0.0071$ <br> $k_2 = 0.0061 \pm 0.0020$ <br> $k_3 = -0.0085 \pm 0.0026$ <br> $k_4 = 0.0029 \pm 0.0022$ | $p_1 = 0.0297$ <br> $p_2 = 0.0017$ <br> $p_3 = 0.0014$ <br> $p_4 = 0.1948$ | 0.2076 |

Response distribution: *Inverse Gaussian*.

Link function: *identity*.

*T or N*: 'Training' (valued 1) or 'No Training' (valued 0) conditions.

*βLFP*: Average LFP beta power during the 2 s before the Go-cue.

*Dur1LFP*: Accumulated LFP beta burst duration during the 2 s before the Go-cue.

*Dur2LFP*: Average LFP beta burst duration during the 2 s before the Go-cue.

*NumLFP*: LFP beta burst number during the 2 s before the Go-cue.

*γLFP*: Average LFP gamma (55–95 Hz) power during the 2 s before the Go-cue.

*αLFP*: Average LFP alpha (8–12 Hz) power during the 2 s before the Go-cue.

*βEEG*: Average EEG beta power during the 2 s before the Go-cue.

*αEEG*: Average EEG alpha (8–12 Hz) power during the 2 s before the Go-cue.

## Neurofeedback training targeting STN beta activity increased tremor

Five out of the twelve participants (nine STN hemispheres) in the study displayed resting tremor during the recording, which enabled us to investigate how volitional suppression of STN beta oscillations affected tremor in Parkinson's disease. The tremor severity, quantified based on the measurements from the tri-axial accelerometer attached to the contralateral hand, increased during the 'Training' condition compared to the 'No Training' condition contralateral to seven out of the tested nine hemispheres (*Figure 6C*, $t_8$ = 3.2589, p = 0.0115). GLME modeling (*Tremor* ~ $k_1 \times T\,or\,N + k_2 \times \beta + k_3 \times \theta + 1|SubID$) confirmed the significant effect of neurofeedback training (*T or N*: $k_1$ = 3.9415 ± 0.4925, p < 0.0001) on increasing tremor. It also indicated that increased tremor band activity (*θ*: $k_3$ = 0.6341 ± 0.0499, p < 0.0001) and reduced beta band activity (*β*: $k_2$ = −0.5971 ± 0.1990, p = 0.0028) in the STN LFPs predicted increased tremor. Overall, the model explained 58.39% of the variance in the tremor power ($R^2$ = 0.5839). When the theta power in the EEG was included in the model, the prediction was not improved (k = −0.1526, p = 0.1103). In addition, a significantly positive correlation was observed between the tremor power and the theta band power in the STN LFP across hemispheres (R = 0.5003, p = 0.034, Pearson's, *Figure 6—figure supplement 2*). There was no significant difference in the tremor severity between 'Training' and 'No Training' conditions when 75% of trials with matched normalized beta power from the two conditions were considered ($t_8$ = −1.1152, p = 0.2971, *Figure 6—figure supplement 1D*). These results suggested that the difference in the experimental condition by itself did not lead to significant difference in the tremor severity between the 'Training' and 'No Training' conditions if the beta power was the same.

## Overnight learning effect of the neurofeedback training

In most EEG-based neurofeedback studies, training sessions are repeated over several separate days (*Engelbregt et al., 2016*; *Schabus et al., 2017*). In this study, four participants (eight hemispheres) repeated the task on two separate, consecutive days. Comparing against Day 1, six out of the eight tested hemispheres showed increased neurofeedback control (indicated by the increased difference in the 'Training' and 'No Training' conditions) on Day 2 (*Figure 7A*). The other two tested hemispheres that had already achieved good neurofeedback control on Day 1 did not further improve on Day 2 (H7 and H8 in *Figure 7A*).

GLME modeling using the difference in the basketball's final position, average beta power, or accumulated beta burst duration between 'Training' and 'No Training' conditions as dependent variable, experimental day (Day 1 or Day 2) as fixed predictor, and a random intercept for each hemisphere confirmed a significant interaction between experimental condition and recording days on the basketball's final position ($k = 0.1497 \pm 0.0372$, $p = 0.0001$), average beta power ($k = -12.56 \pm 3.8987$, $p = 0.0017$), and accumulated beta burst duration ($k = -0.1803 \pm 0.0632$, $p = 0.0051$), suggesting the neurofeedback training on Day 2 was associated with better neurofeedback control and more reduction in the average beta power and accumulated beta burst duration compared to Day 1 (*Figure 7A–C*). There was no significant change in the baseline beta power during rest between Day 1 and Day 2 (*Figure 7D*).

To investigate whether the baseline beta power changes overnight, GLME modeling using the average beta power as dependent variable, experimental condition ('Training' or 'No Training') and experimental day (Day 1 or Day 2) as fixed predictor, and a random intercept for each hemisphere was applied. Apart from the significant interaction between experimental condition and the average beta power ($k = -0.5835$, $p < 0.0001$), the results also confirmed a significant interaction between experimental day and average beta power ($k = -0.1949$, $p = 0.0108$), which could not be explained by the different experimental conditions, suggesting a baseline reduction of the beta power over the two consecutive training days. There was no significant baseline change if we replaced average beta power by accumulated beta burst duration in the model ($k = 0.0041$, $p = 0.8996$).

For the two patients (four hemispheres) who had tremor and repeated the task over two consecutive days, tremor during the 'Training' condition was increased more on Day 2 than Day 1 in all four

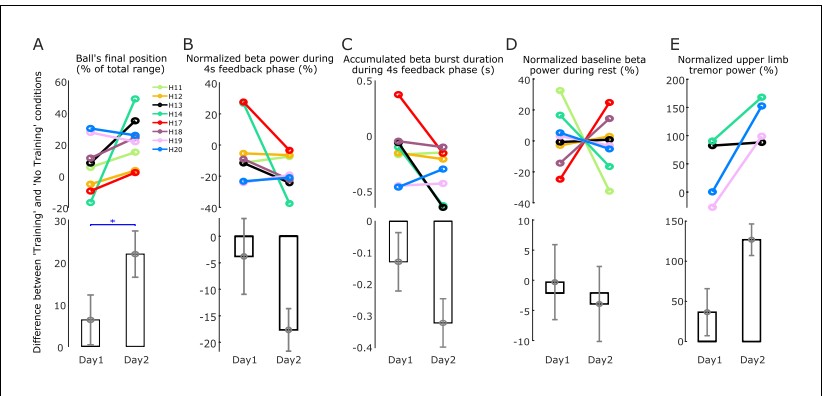

**Figure 7.** Comparison between two training days. (**A**) The difference in the basketball's final vertical position between the 'Training' and 'No Training' conditions, an indication of the neurofeedback control performance, was significantly increased on Day 2 compared to Day 1. (**B**) The reduction in the average normalized beta power in the 'Training' condition compared to the 'No Training' condition was further enhanced on Day 2 compared to Day 1. (**C**) The reduction in the total beta burst duration in the 'Training' condition compared to the 'No Training' condition was further enhanced on Day 2 compared to Day 1. (**D**) There was no significant change in the baseline beta power during rest between Day 1 and Day 2. The baseline beta power was quantified during all the time periods when the participants were at rest throughout the whole experiment session and then normalized by dividing the mean value across 2 days to achieve the percentage change value. (**E**) The increase in the normalized tremor power in the 'Training' condition compared to the 'No Training' condition was also enhanced during Day 2 compared to Day 1. Individual hemispheres and group-averaged data are shown in the upper and lower panels, respectively. Values are presented as mean ± SEM; *p < 0.05 (Wilcoxon signed rank test).

hemispheres (*Figure 7E*). Considering all the individual trials across the two recording days for these hemispheres, GLME modeling using the average tremor power as dependent variable, experimental condition (*T* or *N*: 'Training' or 'No Training'), experimental day (Day: 1 or 2), average beta power ($\beta$), and theta power ($\theta$) in the STN LFP as fixed predictors, and a random intercept for each hemisphere confirmed significant effects for all predictors (*T* or *N*: $k = 4.1901 \pm 0.5696$, p < 0.0001; Day: $k = 3.2611 \pm 0.5477$, p < 0.0001; $\beta$: $k = -0.6253 \pm 0.2073$, p = 0.0027; $\theta$: $k = 0.7016 \pm 0.0487$, p < 0.0001), suggesting the reduced beta and increased theta power in the STN during neurofeedback training on Day 2 associated with the increased tremor.

## Discussion

This is the first study to show that volitional suppression of beta bursts in the STN LFP facilitated by neurofeedback training is able to speed up movement initiation in subsequent cued movement in Parkinsonian patients. This is consistent with previous studies that demonstrate a positive correlation between purposely induced beta-power and reaction time (*Khanna and Carmena, 2017*; *Peles et al., 2020*). We also showed that the suppression of beta was accompanied by an increase in the broad gamma band activity in the STN. Both the reduced beta and increased gamma in the STN LFP before the Go-cue predicted faster reaction time.

### Neurofeedback training for Parkinson's disease

Neurofeedback training aiming to train subjects to self-regulate their neural activity has been proposed to be a promising technique to tune pathological brain activities underlying different diseases (*Ros et al., 2014*).

In the current study, online visual feedback reflected the activity that has been previously related to motor impairment in Parkinson's disease (*Kühn et al., 2006*) – the beta band oscillations in the STN LFPs recorded from the electrode implanted for DBS. We selected a patient-specific beta frequency band that was modulated by voluntary movements and was also enhanced relative to other frequency bands during rest. Our paradigm took into account the temporal dynamics of the signal of interest and reduced the variance and noise in the visual feedback that are not behaviorally relevant, thus allowing Parkinsonian patients to learn to suppress beta bursts within 30 min of training even when off medication. This was accompanied by reduced reaction time in cued movements, which strengthens the link between STN beta, particularly beta bursts, and motor impairment and also suggests that neurofeedback training may help patients develop a strategy to speed up movement initiation.

It should be acknowledged that proper sham control would be required to determine whether observed behavioral and electrophysiological alterations were due to veritable neurofeedback or mediated by other mental strategies (*Thibault et al., 2015*; *Thibault et al., 2016*). Our recent study (*He et al., 2020*) with double-blinded sham control in a similar paradigm targeting the EEG sensorimotor beta activity in young healthy participants showed that veritable neurofeedback had extra effect compared to mental strategies. Thus, considering that externalized patients provide a rare opportunity to understand the response of STN activity to interventions, we did not include a sham condition but only used veritable neurofeedback. Here we argue that veritable neurofeedback may help patients to develop an efficient mental strategy to modulate targeted pathological activities in a short period of time. Our recent study (*He et al., 2020*) suggested that suppression of sensorimotor cortex beta bursts facilitated by neurofeedback training could help improve movement initiation in healthy subjects. The current study suggests that suppression of STN beta bursts facilitated by neurofeedback training also led to a trend of reduced beta over the motor cortex, and reduced beta band coherence between the STN and ipsilateral motor cortex. In addition, it helped improve movement initiation in Parkinson's disease. Even though STN beta is shown to be a more consistent biomarker for bradykinesia in Parkinson's disease, cortical beta oscillation can be measured noninvasively and using cortical beta as neurofeedback signal may make the method more feasible in patients. However, it remains to be tested whether EEG-based neurofeedback training could be used to suppress STN beta bursts and improve movement initiation in Parkinson's disease.

## Broad band gamma activities in STN LFP for Parkinson's disease

In this study, we observed significant increase in the broad band gamma activity accompanied with reduced beta in the STN LFPs during the neurofeedback phase and during the short period of time after the neurofeedback disappeared. In addition, both the reduced beta and increased gamma in the STN LFPs before the Go-cue contributed to the prediction of shorter reaction times. The increase of gamma and reduction of beta band activity in STN have been reported during voluntary movements (*Androulidakis et al., 2007*; *Kempf et al., 2009*; *Brücke et al., 2012*; *Brücke et al., 2013*). The level of gamma increase and beta reduction during the onset of voluntary gripping movements also helps predict gripping force and movement speed (*Tan et al., 2016*; *Lofredi et al., 2018*). In the dopamine-depleted state, movement-related subcortical gamma power significantly decreased (*Kempf et al., 2009*; *Litvak et al., 2012*), particularly during the trials when peak velocity was slower than ON medication (*Lofredi et al., 2018*). These studies suggest that in addition to increased synchrony in the beta band, reduced subcortical gamma signaling in the dopamine-depleted state may also contribute to bradykinesia. The present study shows that Parkinsonian patients were able to purposely increase subcortical gamma band activities. The observed effect in the gamma frequency band may have been mediated by the mental strategy or arousal, since a previous study has shown that STN gamma activity increased during motor imagery and scaled with imagined gripping force (*Fischer et al., 2017*). We also showed that increases in gamma oscillations before the Go-cue predict faster reaction time, over and above the prediction afforded by reduced beta band activities. These results suggest that gamma oscillations may be another important treatment target for Parkinson's disease. Treatments increasing subcortical gamma oscillations, such as medication with levodopa (*Androulidakis et al., 2007*), may also help improve motor initiation.

## Different pathophysiology underlying akinesia-rigidity and tremor in Parkinson's disease

Another important observation in this study is that neurofeedback training targeting beta oscillations may increase tremor, as well as tremor band activities in the STN LFP in tremulous patients. This was not just due to increased cognitive load during the neurofeedback phase since the tremor got worse on Day 2 even though neurofeedback control was improved. Our results are consistent with previous studies showing that, in the presence of tremor, neuronal oscillations at tremor frequency (3–7 Hz) tend to increase in the cortical-basal ganglia-thalamic circuit (*Hirschmann et al., 2013*), whereas beta power (13–30 Hz) and beta band coupling in the motor network are reduced (*Qasim et al., 2016*). Therefore, neurofeedback training targeting beta activity might not help patients with tremor. Such patients might be better served by neurofeedback training focusing on tremor-related oscillations.

## Over-night training sessions

We showed that the patients' ability to modulate their STN beta activity during the neurofeedback phase increased in Day 2 compared to Day 1, even though the baseline beta activities during rest were similar during Day 1 and Day 2. In particular, those patients who did not achieve good neurofeedback control carried on learning and showed significant improvement on Day 2 compared with Day 1. These results suggest that spaced training may facilitate further learning. However, it also remains to be tested if spaced training across multiple sessions would attenuate the connections in the targeted neural network that give rise to synchronization through Hebbian plasticity (*Legenstein et al., 2008*; *Ros et al., 2014*) and whether spaced training can lead to reduced beta synchrony even during rest outside of the neurofeedback task. It would also be interesting to test the effect of neurofeedback training spread out over longer periods as chronic sensing with bidirectional devices becomes more widely available (*Herron et al., 2017*; *Khanna et al., 2017*; *Haddock et al., 2018*; *Houston et al., 2019*).

## Limitations

A within-participant design comparing the 'Training' against the 'No Training' conditions was used in this study to evaluate the effect of neurofeedback training. In a separate study with young healthy participants, we showed that 'veritable feedback' is better than 'sham feedback' in training participants to modulate neural activities even when using similar self-reported mental strategies

(*He et al., 2020*). We did not use 'sham feedback' in the current study because intermixing 'sham feedback' and 'veritable feedback' might have had a negative impact on motivation and might have interfered with learning given the time constraints we had in the patients with externalized electrodes. Therefore, with the current study, we cannot disambiguate whether the observed effects are due to the neurofeedback training or mediated by mental strategy (motor imagery). However, the main results remain valid: Parkinsonian patients can purposely modulate pathological subcortical brain activities, and this modulation led to improved movement initialization. In addition, the more beta band reduction and increase in gamma band activities before the Go-cue predicted faster reaction time.

In summary, we designed a neurofeedback paradigm targeting the neural signal that has previously been shown to be related to bradykinesia and rigidity in Parkinson's disease – beta bursts in the STN. By tailoring the paradigm to the patient-specific beta frequency band and taking into account the temporal dynamics of the signal of interest, the paradigm allowed patients to purposely suppress pathological beta oscillations in the STN within a short training session. The training also led to reduced coupling between the STN and EEG over the motor cortex in the targeted frequency band, as well as to an increase in broad band gamma activity in the STN LFP. Importantly, these changes were accompanied by a reduction in cued reaction time. The results strengthen the link between STN beta oscillations, beta bursts in particular, and motor impairment. Although gamma activity also changed with neurofeedback, multilevel modeling showed that gamma and beta effects independently help predict reaction times. Thus, the results also identify STN gamma activities as an important target for treating motor impairment in Parkinson's disease. The effects of neurofeedback on motor initiation were encouraging, and there was also some indication that the behavioral effects of neurofeedback training might increase over consecutive days. It remains to be seen whether this can translate into a prolonged effect on voluntary motor control, and whether this correlates with clinically meaningful symptom amelioration. It should also be noted that when proposing neurofeedback as a potential therapy, symptom-specific biomarker should be used, and its temporal dynamics need to be taken into account.

## Materials and methods

### Subjects

Twelve Parkinsonian patients (four females), who underwent bilateral implantation of DBS electrodes targeting the motor area of the STN, participated in this study. The DBS leads were temporarily externalized (3–6 days) prior to a second surgery to connect the leads to a pulse generator. The placements of the leads were confirmed by fusion of preoperative MRI and postoperative CT scans. All patients had normal or corrected-to-normal vision and an average age of $62 \pm 8.8$ (range 48–75) years and disease duration of $11 \pm 5.1$ (range 5–20) years. Patients showed good response to dopaminergic medication with mean scores of the Unified Parkinson's Disease Rating Scale (UPDRS) of $45 \pm 13.1$ and $22.9 \pm 9.1$ for medication OFF and ON, respectively. All experiments were conducted with the patients off their dopaminergic medication overnight. The study was approved by the local ethics committees and all patients provided their informed written consent according to the Declaration of Helsinki before the experiments. The clinical details of the patients are summarized in *Table 1*.

### Experimental protocol

The neurofeedback training protocol comprised multiple short trials, similar to what was used in a previous study with healthy young participants (*He et al., 2020*). Each trial consisted of a 2–3 s period during which the patients were instructed to get ready, and a neurofeedback phase lasting 4–8 s followed by a cued motor task 2–3 s after the neurofeedback phase (see Figure A). During the neurofeedback phase, an image of a basketball was presented on a monitor with the vertical position of the basketball indicating the incidence of high amplitude beta bursts quantified in real-time based on the STN LFP measurements. The vertical movement of the basketball was sensitive to the STN beta power calculated within 500 ms long moving windows in real-time. For each update, which occurred every 250 ms (so that windows overlapped), if the calculated beta power was larger than a predefined threshold $T$, the basketball dropped downwards by a fixed distance. The distance of

each drop of the basketball was set so that if the patient was in a resting state, the basketball would drop down to the bottom of the screen within 4–8 s due to spontaneous variations in the power of beta oscillations. If the threshold was not crossed, the ball only moved horizontally on the screen. Thus, the position of the basketball was independent from other variations in beta power that were lower than the threshold used to define beta bursts. This design reduced noise in the visual feedback, and thereby helped participants to gain a sense of agency within a short time period. In the 'Training' condition trials, participants were instructed to try to keep the ball floating at the top of the monitor screen during the neurofeedback phase. The patients were explicitly told that imagining moving their contralateral hand may help to improve the performance but were also encouraged to try different strategies without any real movements. In order to control for effects caused by attending to the moving visual stimulus, participants also performed the task in a 'No Training' condition, in which they were instructed to pay attention to the ball movement and get ready for the Go-cue without having to voluntarily control the position of the ball, though the ball was also moving toward the right as in the 'Training' condition, and the vertical position was controlled by the natural ongoing variations in beta activity.

A Go-cue appeared 2–3 s after the neurofeedback phase to prompt the participants to perform a finger pinch movement. All participants were reminded to avoid any voluntary movements until the Go-cue was presented, and then to pinch a small force meter as fast as possible using their thumbs in response to the Go-cue. The force meter was held on a table by the participant throughout the whole experiment.

Each experimental session consisted of 30 s of rest, a block of 10 trials in the 'Training' condition, and a block of 10 trials in the 'No Training' condition (*Figure 1B*). The instruction for each block was presented for 10 s before the block started. The order of training and no training blocks was randomized in each session. During the 30 s rest period, the power of the selected beta frequency was calculated every 250 ms, and the 75th percentile of the beta power calculated during this 30 s period was then used as the threshold *T* for triggering the vertical movement of the basketball in the following session.

Nine out of the twelve participants completed four sessions of the task separately with both hemispheres and contralateral arms, and the other three participants only completed the task with the dominant hand for the motor task and the contralateral STN. All trials were visually inspected and those with obvious movement artifact during the neurofeedback phase were excluded. Short breaks were provided between sessions, and the recording for each STN lasted for around 30 min. Four patients repeated the same task over two consecutive days with both hemispheres, which allowed us to investigate overnight learning effects.

## Data recording

All recordings in this study were undertaken 3–6 days after the first surgery for bilateral DBS electrodes (Quadripolar Macroelectrode, Model 3389, Medtronic or Vercise Cartesia Directional Lead, Boston Scientific) implantation and prior to the second surgery for connecting the electrodes to the subcutaneous pulse generator. For directional DBS leads, the segmented contacts of levels 2 and 3 were ganged together to make one monopolar channel for the recording. Eight monopolar channels of bilateral STN LFPs and eight monopolar channels of EEG signals covering 'Fz', 'FCz', 'Cz', 'Oz', 'C3', 'C4', 'CP3', and 'CP4' according to the standard 10–20 system were recorded using a TMSi Porti amplifier (TMS International, The Netherlands) at a sampling rate of 2048 Hz. A common average reference was applied automatically to all recorded monopolar signals by the amplifier. The ground electrode was placed on the left forearm. Electromyography (EMG) was simultaneously recorded using the same amplifier from Flexor Carpi Radialis of both arms and the masseter muscle. One tri-axial accelerometer was taped to the back of each hand in order to monitor kinematic movements and any tremor. Generated force in the cued pinch movements was recorded using a pinch meter (P200, Biometrics Ltd). In addition, the real-time positions (X, Y) of the basketball in each trial, which allowed evaluation of the performance of neurofeedback training during the online experiment, and the trigger signals of the paradigm were recorded through an open-source toolkit named Lab Streaming Layer (LSL) (*Kothe, 2014*). The synchronization between different data streams was achieved through LSL and another open-source toolkit named Openvibe (*Renard et al., 2010*). The paradigm used in this study was developed in C++ (Visual Studio 2017, Microsoft) and the online/offline data processing was achieved in Matlab (R2018a, MathWorks, US).

## Selecting the STN LFP channel and the target frequency band

Prior to each experiment, monopolar STN LFPs and EEG data were first recorded during 60 s at rest and during 15 trials of cued finger pinch movements with each hand (*Tan et al., 2015*). The recorded monopolar STN LFPs were re-montaged to bipolar LFPs (through subtraction of adjacent annular or pseudo-annular contacts) prior to analysis. The movement-related power reduction for each bipolar LFP channel contralateral to the performing hand in the beta frequency band (13–30 Hz) was calculated and the bipolar LFP channel with the maximal reduction during movement was selected as the target LFP channel. A 5 Hz frequency band around the frequency showing maximal movement-related modulation ([*f-2, f+2*]) was determined as the individual-specific beta frequency band. The selected bipolar STN LFP channels and the selected frequency band for each STN are listed in *Table 1*. The frequency showing maximal movement-related modulation ranged from 17.4 Hz to 21.4 Hz across all tested STNs and coincided with the peak in the average power spectral density of the selected bipolar channel during rest (*Figure 2A*).

## Estimating beta power in real-time to determine the position of visual feedback online

During the online experiment, the beta power of the selected frequency band was calculated in real-time every 250 ms using a segment of 500 ms data (with 50% overlapping) recorded from the selected bipolar LFP channel. For each segment of 500 ms data, we first applied a mean subtraction followed by a 5–85 Hz band pass filter on the raw data. Next, FFT was applied to calculate the power spectrum of the filtered data and the average power of the selected frequency band was quantified as the beta band power of the current update. At the beginning of each session, data were recorded with the participant resting for 30 s, during which time the beta band power was also updated every 250 ms (119 times). From these values, we selected the 75th percentile as the threshold $T$ for that recording session, which means that when the patient was at rest, their beta power would exceed the threshold 25% of the time (*Tinkhauser et al., 2017a*; *Tinkhauser et al., 2017b*). The threshold was recalculated at the beginning of each session in order to correct for any drift in the average beta power with time spent in the task.

In this paradigm, the position of the basketball was updated every 250 ms, which corresponded to 16–32 updates during the 4–8 s of neurofeedback in each trial. For each update, the displacement of the basketball on the horizontal axis was constant, so the basketball moved from left to right at constant speed. The displacement of the basketball on the vertical axis was related to the beta band power calculated in real-time. When the updated beta power was larger than the threshold $T$, the basketball displayed on the screen dropped downwards by one step. The distance of each step was calibrated, so that the basketball would drop to the bottom of the screen if beta was over the threshold for 25% of the update time points during the feedback phase (4–8 s). Thus, the final vertical position of the basketball in each trial was directly associated with the number of incidences when beta power exceeded the threshold within that time window.

## Offline data analysis

### Visual feedback

The trajectory of the basketball and the final vertical position of the basketball in each individual trial were recorded. The difference between the final vertical positions of the basketball between the 'Training' and 'No Training' conditions indicated the effect of the neurofeedback training. The variations across training days in the differences in the ball's final vertical positions between these two conditions indicated the learning effect induced by neurofeedback training.

### Motor performance

We quantified the reaction time in response to the Go-cue for each trial based on the recorded pinch force. Specifically, the measured force was first low-pass filtered with a 20 Hz cut-off frequency using a fourth order zero-phase digital filter and segmented into 4 s epochs extending between 1 s prior to and 3 s after the Go-cue. We then calculated a threshold to define pinch onset by taking the mean plus three times the standard deviation (SD) of a segment of 500 ms force data before the cue of the pinch task. The time delay between the Go-cue and the time point when the force crossed the determined threshold and sustained for at least 100 ms was taken as the RT of that trial. Force

measurements from individual trials were visually inspected; those trials with obvious artifacts failed to pinch within 2 s after the Go-cue, or with a reaction time smaller than 0.2 s were excluded. Thus, for each of the 21 STN hemispheres we analyzed 44.38 ± 3.88 (mean ± SEM) and 44.57 ± 3.84 trials in the 'Training' and 'No Training' conditions, respectively, resulting in 1868 trials in total across all tested hemispheres.

Hand tremor was monitored by a tri-axial accelerometer attached to the back of each hand. The power in the tremor frequency band (3–7 Hz) was quantified for each axis separately and then averaged across all axes.

## Offline analysis of STN LFP and EEG

The LFPs from the selected STN bipolar channel and EEGs recorded over motor cortex (C3 or C4) were further analyzed off-line with Matlab (v2018a, MathWorks, US). The signals were first band-pass filtered between 0.5 and 100 Hz and notch filtered at 50 Hz using a fourth order zero-phase digital filter. Time-frequency decomposition was obtained by continuous complex Morlet wavelet transformation with a linear frequency scale ranging from 1 to 95 Hz with 1 Hz resolution, and a linearly spaced number (4–8) of cycles across all calculated frequencies. The calculated power of each time point and each frequency was first normalized against the average value quantified across all the time periods when the participants were at rest throughout the whole experiment for that frequency, in order to derive the percentage change. The time courses of beta power percentage changes were separately averaged across trials in the 'Training' and 'No Training' conditions. The average normalized power in the frequency band and time window of interest were calculated for each individual trial for further analysis. In the offline analysis, different beta burst characteristics (accumulated duration, average duration, and number of bursts) during the first 4 s of the neurofeedback phase were recalculated as in *Tinkhauser et al., 2017a*. In order to investigate whether there would be a similar impact of neurofeedback training on the power and bursts in other non-targeted frequency bands, for each hemisphere, we repeated the power and burst characteristics calculation and analyses in two other frequency bands which were not overlapping with the selected 5 Hz beta band by shifting the center frequency band by 8 Hz down and up, to give 'Beta−8 Hz' and 'Beta+8 Hz' frequency bands.

The connectivity between the STN LFP and ipsilateral motor cortex EEG was evaluated using the phase synchrony index (PSI, *Equation 1*; *Lachaux et al., 2000*) and spectral coherence (Coh, *Equation 2*; *Lachaux et al., 1999*) calculated based on the time-frequency decomposition results after complex Morlet transformation, and compared between experimental conditions ('Training' or 'No Training').

$$PSI = \left| n^{-1} \sum_{t=1}^{n} e^{i\left(\varphi_{lfp}^{t} - \varphi_{eeg}^{t}\right)} \right| \tag{1}$$

$$Coh = \frac{\left| n^{-1} \sum_{t=1}^{n} |m_{lfp}^{t}||m_{eeg}^{t}| e^{i\left(\varphi_{lfp}^{t} - \varphi_{eeg}^{t}\right)} \right|^{2}}{\left( n^{-1} \sum_{t=1}^{n} |m_{lfp}^{t}|^{2} \right) \left( n^{-1} \sum_{t=1}^{n} |m_{eeg}^{t}|^{2} \right)} \tag{2}$$

where $n$ indicates the total time points in each trial (4 s), $\varphi_{lfp}^{t}$ and $\varphi_{eeg}^{t}$ indicate the phase values of the selected LFP and EEG signals at time point $t$, and $m_{lfp}^{t}$ and $m_{eeg}^{t}$ indicate the amplitude values of the selected LFP and EEG signals at time point $t$, respectively.

## Generalized linear mixed effects modeling

GLME modeling (Matlab function *fitglme*) was used to assess the trial-to-trial within-subject relationship between different measurements, and how they were changed by neurofeedback training. Apart from transforming the dependent variable to eliminate the deviation from normality distribution, GLME also allows researchers to select a theoretical distribution that matches the properties of the dependent variable (*Lo and Andrews, 2015*). For example, the measured RT is skewed and closer to an Inverse Gaussian distribution instead of a normal Gaussian distribution, thus an Inverse Gaussian distribution was selected in the models using RT as dependent variable. When applying

GLME modeling, data from all valid individual trials from all tested hemispheres were considered, and the average power (10log10 transferred to dB) was used when applicable. The slope(s) between the predictor(s) and the dependent variable were set to be fixed across all hemispheres; a random intercept was set to vary by hemisphere. The details of the models were described together with the results.

## Statistical analysis

Paired $t$-tests (Matlab function $t$-test) or nonparametric Wilcoxon signed-rank test (Matlab function $signrank$), depending on whether the normal distribution assumption was satisfied, was used to evaluate the effect of the experimental condition ('Training' and 'No Training') on neurofeedback task performance, the motor task reaction time, tremor severity, and neural activities measured in STN LFPs and EEGs. The normal distribution assumption was tested using Anderson–Darling test (Matlab function $adtest$) (*Anderson and Darling, 1952*). Multiple comparisons applied to different measurements were corrected using Bonferroni correction.

When GLME modeling was used, the estimated fixed effect coefficient ($k$), which indicates the potential positive or negative correlation between the predictor and the dependent variable, the corresponding t-statistic p-value, and $R^2$ were reported.

## Acknowledgements

We thank the patients who participated for making this study possible.

## Additional information

### Competing interests

Keyoumars Ashkan: has received educational grants from Medtronic and Abbott. Peter Brown: is a consultant for Medtronic. The other authors declare that no competing interests exist.

### Funding

| Funder | Grant reference number | Author |
| --- | --- | --- |
| Medical Research Council | MR/P012272/1 | Shenghong He<br>Huiling Tan |
| Medical Research Council | MC_UU_12024/1 | Flavie Torrecillos<br>Gerd Tinkhauser<br>Petra Fischer<br>Alek Pogosyan<br>Peter Brown |
| National Institute for Health Research | Oxford Biomedical Research Centre | Shenghong He<br>Abteen Mostofi<br>Emilie Syed<br>Flavie Torrecillos<br>Gerd Tinkhauser<br>Petra Fischer<br>Alek Pogosyan<br>Peter Brown<br>Huiling Tan |
| Rosetrees Trust | | Shenghong He<br>Huiling Tan |

The funders had no role in study design, data collection and interpretation, or the decision to submit the work for publication.

### Author contributions

Shenghong He, Data curation, Software, Formal analysis, Validation, Investigation, Visualization, Methodology, Writing - original draft, Writing - review and editing; Abteen Mostofi, Resources, Data curation, Validation, Writing - review and editing; Emilie Syed, Conceptualization, Validation, Investigation, Writing - review and editing; Flavie Torrecillos, Gerd Tinkhauser, Petra Fischer, Data curation,

Validation, Writing - review and editing; Alek Pogosyan, Software, Validation; Harutomo Hasegawa, Resources, Data curation, Validation; Yuanqing Li, Keyoumars Ashkan, Resources, Validation, Writing - review and editing; Erlick Pereira, Conceptualization, Resources, Validation, Investigation, Writing - review and editing; Peter Brown, Conceptualization, Resources, Supervision, Funding acquisition, Validation, Investigation, Project administration, Writing - review and editing; Huiling Tan, Conceptualization, Resources, Data curation, Supervision, Funding acquisition, Validation, Investigation, Methodology, Project administration, Writing - review and editing

### Author ORCIDs
Shenghong He https://orcid.org/0000-0002-5269-1902
Petra Fischer http://orcid.org/0000-0001-5585-8977
Peter Brown https://orcid.org/0000-0002-5201-3044
Huiling Tan https://orcid.org/0000-0001-8038-3029

### Ethics
Human subjects: Informed consent and consent to publish was obtained from patients before they took part in the study, which was approved by Oxfordshire Research Ethics Committee, reference number 18/SC/0006.

### Decision letter and Author response
Decision letter https://doi.org/10.7554/eLife.60979.sa1
Author response https://doi.org/10.7554/eLife.60979.sa2

## Additional files

### Supplementary files
• Source code 1. Source data and codes for *Figures 2*, *3*, *4*, *5*, *6*, *7*, all supplement figures, and Table 2.

• Transparent reporting form

### Data availability
Source data and codes for generating Figures 2-7, all supplement figures, and Table 2 have been provided.

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
