## [Decision Letter]

**Acceptance summary:**

This is an innovative study where patients with Parkinson's disease and deep brain stimulation devices use real-time feedback in order to reduce their subthalamic nucleus (STN) beta band power. The training condition (where beta band power is lowered) precedes faster reaction times in a pinching task. This study extends the use of sequential neurofeedback-motor tasks that have been used in animal models and healthy participants to a patient population, and demonstrates a link between lowering STN beta, a biomarker thought to reflect the bradykinesia and rigidity state, and improved movement initiation. Further, the study shows that patients improve their ability to lower STN beta with a 2nd day of practice, and that lowering STN beta increases tremor, a novel finding and important consideration for the use of neurofeedback paradigms to improve motor symptoms in parkinsonian patients.

**Decision letter after peer review:**

Thank you for submitting your article "Subthalamic beta targeted neurofeedback training speeds up movement initiation but increases tremor in Parkinsonian" for consideration by *eLife*. Your article has been reviewed by three peer reviewers, and the evaluation has been overseen by Preeya Khanna as the Guest Reviewing Editor and Richard Ivry as the Senior Editor. The following individuals involved in review of your submission have agreed to reveal their identity: Tomas Ros (Reviewer #2); Preeya Khanna (Reviewer #3).

The reviewers have discussed the reviews with one another and the Reviewing Editor has drafted this decision to help you prepare a revised submission.

Summary:

Overall, the authors present a study in patients with Parkinson's disease and recent DBS implants of using subthalamic nucleus (STN) beta neurofeedback in a sequential neurofeedback-behavior task setup. They show that when participants are training with neurofeedback, they are able to reduce STN beta power, burst duration, and number of burst and increase STN gamma power. The "training" condition with NF precedes faster reaction times, and in a subset of cases, higher tremor. Finally, the authors show that patients can improve their NF performance on a 2nd consecutive day of training. Overall the reviewers agreed that this was an interesting and innovative study.

1) The common major concern was the lack of controls performed in order to identify whether the reduced reaction times (and increased tremor in some patients) were due to patients engaging in a more cognitively demanding task that required higher alertness ("training") compared to a less cognitively demanding task ("no training") or whether the lower reaction time (and increased tremor) was due to reductions in beta power in STN. Supporting this concern is the finding that when including a training vs. non-training variable into the regression used to predict RT, that the variable remained significant even in the presence of the beta power variable. This shows that beta modulation does not fully account for the effect of RT, and subjects may be doing something else to improve their RT in the training blocks. The authors are very forthcoming with their lack of control, and that's appreciated! Since we appreciate the difficulties in acquiring patient data in the current pandemic environment, we are wondering if the authors can leverage their current data to demonstrate that there are not significant differences in RT distribution if they select trials from the training and no-training conditions that have similar beta power distributions.

2) Even if the suggested above analysis is able to be completed, the reviewers would like to see many claims (including the title) in the study re-phrased to reflect the ambiguity of whether the neurofeedback task alone or the reduction in beta power resulted in RT/tremor changes. Examples of these phrases and edits to the claims include:

i) "Subthalamic beta targeted neurofeedback training speeds up movement initiation but increases tremor in Parkinsonian" to "Performance of a NF beta suppression task predicts faster movement initiation and increases in tremor in Parkinson's Disease compared to a passive observation task"

ii) "We developed a neurofeedback paradigm targeting beta bursts to investigate whether volitional suppression of STN beta can improve motor performance in Parkinson's disease." to "We developed a neurofeedback paradigm targeting beta bursts and investigated whether performance of the NF task could improve motor initiation in PD"

iii) "Successful volitional beta suppression was associated with an amplification of Parkinsonian tremor, suggesting a role of STN beta in tremor." to “Successful NF task performance was associated with STN beta suppression and amplification of PD tremor.”

iv) "Neurofeedback training reduced beta oscillations in STN LFPs and reduced beta band synchrony between the conditioned STN and ipsilateral motor cortex", same claim but add "compared to a passive observation task" to the end of the sentence.

3) Presentation of more raw data is needed as almost all plots include only pre-processed data. For example, for Figure 3B, it would be helpful to see raw EMG traces, not just bar plots. It is hard to know what 20uV means, and since we are only told there is no sig. difference between the training and no-training it could mean there's some ongoing EMG in both conditions. Either way, would be helpful to see some example trials. Similarly, it would be helpful to present raw trials and subject average time-traces of the pinch force sensor. Right now, we only know that RT changes, but would be nice to see the traces to see if other statistics about the movement onset change (max force, slope of force onset, etc.).

4) Please report R2 (explained variance) values in Table 2. Significance is fine, but nice to also know how much RT variance is being explained by each of the models.

5) The authors refer interchangeably to beta bursts and beta power. They are obviously related, but please clarify that the average beta power was used as a neurofeedback signal (not burst counts).

6) Did neurofeedback reduce mostly long (“pathological”) bursts or also short (“physiologic”) bursts?

7) One major omission in this manuscript is some additional figures (LFP and EEG) of the theta frequency (4-7 Hz) which the authors repeatedly relate to underlie tremor. Please add this for completeness, including a plot of the regression with the accelerometer tremor data.

8) Please discuss the relationship of the current study to the EEG study of (He et al., 2020) and given your observations, the translational potential of using noninvasive EEG neurofeedback as a therapy for Parkinson's in lieu of DBS electrodes.

9) It strikes me as peculiar that the top left inset in Figure 4A Average Power (dB) has a large t-value of t = -4.769 but that visually there is an almost complete overlap in the error bars (compared to the other plots). How may this large effect size be explained given the large errors ? I realize this is a paired t-test but I would recommend to re-verify your analyses as a cautious double check.

10) Given the overnight learning effect of the neurofeedback training, it might be interesting to also discuss the potential neuroplastic mechanisms behind this effect, e.g. Hebbian plasticity (see Ros et al., 2014).

Revisions expected in follow-up work:

When data collection becomes more feasible, we would like to see the authors perform an appropriate control study in either the same or separate cohort of patients. This control could involve NF modulation to increase STN beta (and show an increase in RT) or could involve NF modulation of a difference frequency band (and show no change in RT). These experiments could be summarized in a preprint on bioRxiv or medRxiv, or if appropriate, as a Research Advance in *eLife*, either of which would be linked to the original paper.

---

## [Author Response]

Revisions for this paper:1) The common major concern was the lack of controls performed in order to identify whether the reduced reaction times (and increased tremor in some patients) were due to patients engaging in a more cognitively demanding task that required higher alertness ("training") compared to a less cognitively demanding task ("no training") or whether the lower reaction time (and increased tremor) was due to reductions in beta power in STN. Supporting this concern is the finding that when including a training vs. non-training variable into the regression used to predict RT, that the variable remained significant even in the presence of the beta power variable. This shows that beta modulation does not fully account for the effect of RT, and subjects may be doing something else to improve their RT in the training blocks. The authors are very forthcoming with their lack of control, and that's appreciated! Since we appreciate the difficulties in acquiring patient data in the current pandemic environment, we are wondering if the authors can leverage their current data to demonstrate that there are not significant differences in RT distribution if they select trials from the training and no-training conditions that have similar beta power distributions.

We thank the reviewers for the thoughtful comment. According to the reviewers’ suggestion, we have now selected 75% of trials from the “Training” and “No Training” conditions for each hemisphere, respectively, that have similar beta power distributions. We then compared the reaction time, gamma power, and tremor severity of these trials. As shown in Figure 6—figure supplement 1, there were no significant differences in the RT, gamma power, and tremor severity between “Training” and “No Training” conditions between these trails from different experimental conditions but with matched beta power. These results suggest that the difference in the experimental condition by itself may not lead to a difference in the RT, gamma power, and tremor severity. Thus, beta modulation during neurofeedback training does contribute to the changes in RT, even though some other factors (e.g., cognitive requirement) may also contribute to the observed difference in the RT between the “Training” and “No Training” conditions, as the reviewers pointed out and as the GLME modeling applied to all trials from both conditions revealed. Please see the new Figure 6—figure supplement 1 and revised text:

Results subsection “Neurofeedback training improved reaction time in subsequently cued movements”

“The significant negative k1 showed that there was an effect of “Training” in reducing the reaction time which cannot be explained by changes in the beta or gamma band power. The positive sign of k2 and negative sign of k3 indicate that reduced STN beta band power and increased gamma band power over the 2 s before the Go cue predicted faster reaction time. In addition, we selected a subgroup (75%) of trials from the “Training” and “No Training” conditions that have similar normalized beta power (Figure 6—figure supplement 1A), and tested the differences in reaction time and normalized gamma power. The results showed no significant difference in the RT (*t*_20_ = -0.4374, *p* = 0.6665, Figure 6**—**figure supplement 1B) nor in the normalized gamma power (z = -0.8168, *p* = 0.4140, Figure 6—figure supplement 1C) between the selected trials from the “Training” and “No Training” conditions but with matched normalized beta power. Overall these analyses suggest that beta modulation during neurofeedback training does contribute to the changes in RT, even though other condition factors (e.g., cognitive requirement) may also contribute to the observed difference in the RT between the “Training” and “No Training” conditions.”

Subsection “Neurofeedback training targeting STN beta activity increased tremor”

“There was no significant difference in the tremor severity between “Training” and “No Training” conditions when 75% of trials with matched normalized beta power from the two conditions were considered (*t*_8_ = -1.1152, *p* = 0.2971, Figure 6**—**figure supplement 1D). These results suggested that the difference in the experimental condition by itself did not lead to significant difference in the tremor severity between the “Training” and “No Training” conditions if the beta power was the same.”

2) Even if the suggested above analysis is able to be completed, the reviewers would like to see many claims (including the title) in the study re-phrased to reflect the ambiguity of whether the neurofeedback task alone or the reduction in beta power resulted in RT/tremor changes. Examples of these phrases and edits to the claims include:i) "Subthalamic beta targeted neurofeedback training speeds up movement initiation but increases tremor in Parkinsonian" to "Performance of a NF beta suppression task predicts faster movement initiation and increases in tremor in Parkinson's Disease compared to a passive observation task"ii) "We developed a neurofeedback paradigm targeting beta bursts to investigate whether volitional suppression of STN beta can improve motor performance in Parkinson's disease." to "We developed a neurofeedback paradigm targeting beta bursts and investigated whether performance of the NF task could improve motor initiation in PD"iii) "Successful volitional beta suppression was associated with an amplification of Parkinsonian tremor, suggesting a role of STN beta in tremor." to “Successful NF task performance was associated with STN beta suppression and amplification of PD tremor.”iv) "Neurofeedback training reduced beta oscillations in STN LFPs and reduced beta band synchrony between the conditioned STN and ipsilateral motor cortex", same claim but add "compared to a passive observation task" to the end of the sentence.

We thank the reviewers’ rigorous suggestions. These claims now have been re-phrased accordingly. Please see the revised sentences:

Title

i) We have a limit in the number of characters (120) in the title, therefore we kept the original title “Subthalamic beta targeted neurofeedback speeds up movement initiation but increases tremor in Parkinsonian patients” but has added “compared to passive observation” in the Abstract.

Abstract

ii) “We developed a neurofeedback paradigm targeting STN beta bursts and investigated whether neurofeedback training could improve motor initiation in Parkinson’s disease compared to passive observation.”

iii) “However, in Parkinsonian patients with pre-existing symptoms of tremor, successful volitional beta suppression was associated with an amplification of tremor which correlated with theta band activity in STN LFPs, suggesting an additional cross-frequency interaction between STN beta and theta activities.”

Results

iv) “Neurofeedback training reduced beta oscillations in STN LFPs and reduced beta band synchrony between the conditioned STN and ipsilateral motor cortex compared to a passive observation task”

3) Presentation of more raw data is needed as almost all plots include only pre-processed data. For example, for Figure 3B, it would be helpful to see raw EMG traces, not just bar plots. It is hard to know what 20uV means, and since we are only told there is no sig. difference between the training and no-training it could mean there's some ongoing EMG in both conditions. Either way, would be helpful to see some example trials. Similarly, it would be helpful to present raw trials and subject average time-traces of the pinch force sensor. Right now, we only know that RT changes, but would be nice to see the traces to see if other statistics about the movement onset change (max force, slope of force onset, etc.).

We have now added the averaged EMG trace during the 2 s pre-feedback and 4 s feedback phase for each hemisphere in each condition in the revised Figure 3B, and have added an example of the recorded left-hand pinch force from P12 in Figure 6B. Please see the revised and new added Figures 3B and 6B and the corresponding text:

Results subsection “Neurofeedback training improved reaction time in subsequently cued movements”

“The reaction time in response to the Go cue was significantly reduced in the “Training” condition compared with the “No Training” condition (487.4 ± 29.7 ms compared to 510.9 ± 32.3 ms, t_20_ = -2.7518, p = 0.0123, paired *t* test, Figure 6A). Figure 6B shows an example of the recorded left-hand pinch force in the “Training” and “No Training” conditions from Patient 12.”

4) Please report R2 (explained variance) values in Table 2. Significance is fine, but nice to also know how much RT variance is being explained by each of the models.

We followed this suggestion and have now added the R^2^ for each tested model in the revised Table 2. Overall, around 20% of the variance in the reaction time was being explained by the models. When the “Training” and “No Training” conditions were considered separately in the model, around 24% and 19% of the variance in the reaction time were being explained by the models in the “Training” and “No Training” conditions, respectively. Please see the revised Table 2 and the revised text:

Results subsection “Neurofeedback training improved reaction time in subsequently cued movements”

“Overall, around 20% of the variance in the reaction time was being explained by the model (Model 6, R^2^ = 0.2072, Table 2). The significant negative k1 showed that there was an effect of “Training” in reducing the reaction time which cannot be explained by changes in the beta or gamma band power. The positive sign of k2 and negative sign of k3 indicate that reduced STN beta band power and increased gamma band power over the 2 s before the Go cue predicted faster reaction time.”

5) The authors refer interchangeably to beta bursts and beta power. They are obviously related, but please clarify that the average beta power was used as a neurofeedback signal (not burst counts).

We apologize for the unclear statements regarding this issue. Indeed, the average beta power over each 500 ms was used as a neurofeedback signal to control the basketball movement. We have now clarified this as:

Results subsection “Neurofeedback control was achieved within one day of training”

“The bipolar LFP channel and the peak frequency bands (5 Hz width) with the largest movement-related changes between 13-30Hz were selected to drive the visual feedback for each hemisphere (Figure 2). Specifically, the average power in the selected beta frequency band over each 500 ms time window was used as a neurofeedback signal to control the vertical position of the basketball. In real time, we assumed that a beta burst was detected when the average beta power within the past 500 ms time window exceeded a pre-defined threshold, which would result in a drop of the basketball.”

Materials and methods subsection “Offline analysis of STN-LFP and EEG”

“In the offline analysis, different beta burst characteristics (accumulated duration, average duration, and number of bursts) during the first four seconds of the neurofeedback phase were re-calculated as in Tinkhauser et al., 2017a.”

6) Did neurofeedback reduce mostly long (“pathological”) bursts or also short (“physiologic”) bursts?

According to the reviewers’ comment, we have now compared the distribution profiles of the beta bursts with different durations during the 4s feedback phase between “Training” and “No Training” conditions. As shown in Figure 4—figure supplement 1, the neurofeedback training was associated with an overall reduction of beta bursts of different durations, which was consistent with the results in Figure 4B-D showing a significant reduction of accumulated burst duration, average burst duration, and burst number in “Training” condition compared with “No Training” condition. The difference was more pronounced for bursts with durations longer than 400 ms. Please see the new added Figure 4—figure supplement 1 and the corresponding text:

Results subsection “Neurofeedback training reduced beta oscillations in STN LFPs and reduced beta band synchrony between the conditioned STN and ipsilateral motor cortex compared to a passive observation task”

“The neurofeedback training also led to reduced accumulated beta burst duration in the STN LFPs determined as percentage of time with beta amplitude being over the predefined threshold (*t*_20_ = -4.7415, *p* = 0.0001, 17.40 ± 1.44 % compared to 22.43 ± 1.85%, mean ± SEM, Figure 4B), a reduced average burst duration (*t*_20_ = -3.9428, *p*=0.0008, 319.6 ± 19.3 ms compared to 377.2 ± 21.5 ms, Figure 4C), and a reduced number of bursts per second (*t*_20_ = -4.8536, *p* = 0.0001, 0.446 ± 0.030 compared to 0.531 ± 0.033, Figure 4D). The bursts with durations longer than 400 ms were reduced more consistently compared with the shorter bursts (Figure 4—figure supplement 1).”

7) One major omission in this manuscript is some additional figures (LFP and EEG) of the theta frequency (4-7 Hz) which the authors repeatedly relate to underlie tremor. Please add this for completeness, including a plot of the regression with the accelerometer tremor data.

Thank you for the comment. According to your suggestion, we have now added a plot showing the correlation between the tremor power quantified from accelerometer measurements and the theta power in the STN LFP during the 4s feedback phase for those 9 hemispheres, which displayed contralateral tremor during the experiment. As shown in Figure 6—figure supplement 2, the STN LFP theta power was positively correlated with the accelerometer tremor band power across hemispheres (*R* = 0.5003, *p* = 0.0344, Pearson’s). There was no correlation between the theta power in the EEG and the accelerometer tremor power (*R* = -0.1596, *p* = 0.5269, Pearson’s). Please see the new added Figure 6—figure supplement 2 and text:

Results subsection “Neurofeedback training targeting STN beta activity increased tremor”

“GLME modeling (Tremor∼k1*TorN+k2*β+k3*θ+1|SubID) confirmed the significant effect of neurofeedback training (TorN:k1= 3.9415 ± 0.4925, *p* < 0.0001) on increasing tremor. It also indicated that increased tremor band activity (θ:k3= 0.6341 ± 0.0499, *p* < 0.0001) and reduced beta band activity (β:k2= -0.5971 ± 0.1990, *p* = 0.0028) in the STN LFPs predicted increased tremor. Overall, the model explained 58.39 % of the variance in the tremor power (R2 = 0.5839). When the theta power in the EEG was included in the model, the prediction was not improved (*k* = -0.1526, *p* = 0.1103). In addition, a significantly positive correlation was observed between the tremor power and the theta band power in the STN LFP across hemispheres (R = 0.5003, p = 0.034, Pearson’s, Figure 6—figure supplement 2).”

8) Please discuss the relationship of the current study to the EEG study of (He et al., 2020) and given your observations, the translational potential of using noninvasive EEG neurofeedback as a therapy for Parkinson's in lieu of DBS electrodes.

We followed this suggestion and have now added the following sentences in the Discussion.

Discussion subsection “Neurofeedback training for Parkinson’s disease”

“Our recent study (He et al., 2020) suggested that suppression of sensorimotor cortex beta bursts facilitated by neurofeedback training could help improve movement initiation in healthy subjects. The current study suggests that suppression of STN beta bursts facilitated by neurofeedback training also led to a trend of reduced beta over the motor cortex, and reduced beta band coherence between the STN and ipsilateral motor cortex. In addition, it also helped improve movement initiation in Parkinson’s disease. Even though STN beta is shown to be a more consistent biomarker for bradykinesia in Parkinson’s disease, cortical beta oscillation can be measured non-invasively and using cortical beta as neurofeedback signal may make the method more feasible in patients. However, it remains to be tested whether EEG-based neurofeedback training could be used to suppress STN beta bursts and improve movement initiation in Parkinson’s disease.”

9) It strikes me as peculiar that the top left inset in Figure 4A Average Power (dB) has a large t-value of t = -4.769 but that visually there is an almost complete overlap in the error bars (compared to the other plots). How may this large effect size be explained given the large errors ? I realize this is a paired t-test but I would recommend to re-verify your analyses as a cautious double check.

We thank the reviewers for the comment. Please note that the power in the previous Figure 4 were raw power in log scale (dB). With paired t-test, a large t-value was achieved because for most hemispheres, the values in “Training” condition were consistently smaller than “No Training” condition. The almost overlapped error bars were caused by the large variance in the power amplitude in the recorded signal across hemispheres. We realised that plotting the log transformed power was misleading, therefore, we have now changed the log transformed power into percentage change by dividing the average power of different frequency band during the neurofeedback phase by the average of each frequency band across all resting periods throughout the whole experiment session. All the t-tests are now applied on the normalised percentage change. Most results are similar to when log transformed power were used as in the previous version. The only change is the effect of “Training” on the beta band modulation in the ipsilateral motor cortex measured using EEG was no longer significant after multiple comparison correction. Please see the revised figure and text:

Results subsection “Neurofeedback training reduced beta oscillations in STN LFPs and reduced beta band synchrony between the conditioned STN and ipsilateral motor cortex compared to a passive observation task”

“A paired *t* test confirmed a significant effect of neurofeedback in facilitating beta suppression in terms of the average normalized power in the selected beta bands (*t*_20_ = -3.6975, *p* = 0.0014) (Figure 4A). The difference in the normalized beta power between the “Training” and “No Training” conditions correlated positively with the percentage change in the beta power during real movement (*r* = 0.5896, *p* = 0.0057, Pearson’s correlation, Figure 3D).”

“Although there was a trend of reduction in the average normalized beta power and beta burst characteristics in the EEG recorded over the ipsilateral motor cortex, the changes were not significant or did not survive multiple comparison correction (Figure 4D-H).”

Materials and methods subsection “Offline data analysis”

“The calculated power of each time point and each frequency was first normalised against the average value quantified across all the time periods when the participants were at rest throughout the whole experiment for that frequency, in order to derive the percentage change. The time courses of beta power percentage changes were separately averaged across trials in the “Training” and “No Training” conditions. The average normalized power in the frequency band and time window of interest were calculated for each individual trial for further analysis.”

10) Given the overnight learning effect of the neurofeedback training, it might be interesting to also discuss the potential neuroplastic mechanisms behind this effect, e.g. Hebbian plasticity (see Ros et al., 2014).

We thank the reviewers for the thoughtful comment. Our results show that the base line beta activities during rest were similar during Day 1 and Day 2, but the ability to modulate this activity during the neurofeedback phase increased in Day 2. Hebbian plasticity has now been mentioned and discussed:

Discussion subsection “Over-night training sessions”

“We showed that the patients’ ability to modulate their STN beta activity during the neurofeedback phase increased in Day 2 compared to Day 1, even though the baseline beta activities during rest were similar during Day 1 and Day 2. In particular, those patients who did not achieve good neurofeedback control carried on learning and showed significant improvement on Day 2 compared with Day 1. These results suggest that spaced training may facilitate further learning. However, it also remains to be tested if spaced training across multiple sessions would attenuate the connections in the targeted neural network that give rise to synchronization through Hebbian plasticity (Legenstein et al., 2008; Ros et al., 2014) and whether spaced training can lead to reduced beta synchrony even during rest outside of the neurofeedback task.”

Revisions expected in follow-up work:When data collection becomes more feasible, we would like to see the authors perform an appropriate control study in either the same or separate cohort of patients. This control could involve NF modulation to increase STN beta (and show an increase in RT) or could involve NF modulation of a difference frequency band (and show no change in RT). These experiments could be summarized in a preprint on bioRxiv or medRxiv, or if appropriate, as a Research Advance in eLife, either of which would be linked to the original paper.

We agree with the reviewers that it would be interesting to see if an appropriate control study could support the findings in the current study. The authors are glad to do this once data collection becomes more feasible.